

# Obesity phenotypes and their relationships with atrial fibrillation

Szu-Ying Tsai[1,2], Hsin-Hao Chen[3,6,8], Hsin-Yin Hsu[2], Ming-Chieh Tsai[4,5], Le-Yin Hsu[5], Lee-Ching Hwang[2,6], Kuo-Liong Chien[5,7], Chien-Ju Lin[3] and Tzu-Lin Yeh[3,5]

[1] Department of Family Medicine, Taipei City Hospital, Zhongxing Branch, Taipei City, Taiwan
[2] Department of Family Medicine, Taipei MacKay Memorial Hospital, Taipei City, Taiwan
[3] Department of Family Medicine, Hsinchu MacKay Memorial Hospital, Hsinchu City, Taiwan
[4] Division of Endocrinology and Metabolism, Department of Internal Medicine, Taipei MacKay Memorial Hospital, Taipei City, Taiwan
[5] Institute of Epidemiology and Preventive Medicine, College of Public Health, National Taiwan University, Taipei City, Taiwan
[6] Department of Medicine, MacKay Medical College, New Taipei City, Taiwan
[7] Department of Internal Medicine, National Taiwan University Hospital, Taipei City, Taiwan
[8] MacKay Junior College of Medicine, Nursing, and Management, Taipei, Taiwan

## ABSTRACT

**Background:** This study assessed the associations of metabolic obesity phenotypes with the risk of atrial fibrillation (Afib).

**Methods:** This prospective cohort study categorized Taiwanese adults according to their body mass index (BMI) and metabolic health status at baseline. We assigned the participants to the underweight (BMI < 18.5 kg/m$^2$), normal weight (BMI = 18.5–23.9 kg/m$^2$), and overweight/obesity groups (BMI ≥ 24 kg/m$^2$). Metabolically healthy was defined as absence of hypertension, diabetes, and hyperlipidemia and the presence of healthy metabolic profiles.

**Results:** In total, 5,742 adults were included. During a median follow-up of 13.7 years, 148 patients developed Afib. Compared to the metabolically healthy normal weight group, the risk of Afib was significantly higher than those in the metabolically unhealthy overweight/obesity (hazard ratio = 2.20, 95% confidence interval [1.12–4.33]) and metabolically unhealthy normal weight groups (HR = 2.64, 95% CI [1.34–5.17]). Additionally, the point estimate suggested a 1.97-fold greater risk among the metabolically healthy overweight/obesity group, although this difference was not significant given the wide confidence interval (HR = 1.97, 95% CI [0.80–4.86]).

**Conclusion:** Our results demonstrated the relationships of metabolic health and weight regarding the risk of Afib in Taiwanese adults. The Afib risk among metabolic and obesity phenotypes is associated with a metabolically unhealthy status. A trend toward a higher Afib risk with obesity among metabolically healthy subjects was observed. However, the result was not robust and it still suggested further study.

Corresponding authors
Chien-Ju Lin, tttyii213@gmail.com
Tzu-Lin Yeh, 5767@mmh.org.tw

## INTRODUCTION

Atrial fibrillation (Afib) is one of the most common sustained cardiac arrhythmias globally. The estimated prevalence varies among different regions from 1 to 4% in western countries and from 0.49% to 1.9% in Asia (*Zulkifly, Lip & Lane, 2018*). Afib is an important disease because it is associated with increased risks of stroke, myocardial infarction, heart failure, dementia, and chronic kidney disease and a higher risk of mortality (*Chugh et al., 2014*).

Obesity is a known risk factor of Afib (*Frost, Hune & Vestergaard, 2005*; *Lavie et al., 2017*; *Wanahita et al., 2008*; *Wang et al., 2004*). However, individuals with obesity are highly heterogeneous. People with obesity can be categorized as having metabolically unhealthy obesity (MUO) or metabolically healthy obesity (MHO) (*Lavie et al., 2018*) according to their metabolic profiles, and most studies involving this issue defined metabolic status using different metabolic syndrome (MetS) criteria (*De Lorenzo et al., 2018*). Various studies demonstrated that a metabolic unhealthy status is associated with an increased risk of Afib (*Kim et al., 2018*; *Kwon et al., 2019*; *Watanabe et al., 2008*). Each MetS component is independently associated with an increased incidence of Afib (*Kwon et al., 2019*; *Vyssoulis et al., 2013*), and the trend of developing Afib increases with increasing numbers of MetS components present (*Watanabe et al., 2008*). However, we know little about the health outcomes of individuals with MHO. Only three published studies investigated the relationship between MHO and Afib, all of which revealed an increased Afib risk in the MHO group (*Feng et al., 2019*; *Lee et al., 2017*; *Nystrom et al., 2015*). However, these studies adopted inconsistent definitions (*Alberti, Zimmet & Shaw, 2006*; *Alberti et al., 2009*; *Expert Panel on Detection E & Treatment of High Blood Cholesterol in A, 2001*) of a metabolically healthy status. Moreover, these definitions contain different metabolic components, raising concerns regarding whether the populations sufficiently represented people with MHO or only people with obesity and fewer metabolic abnormalities.

Because the relationship between MHO and Afib remains uncertain, this research gap is worthy of study. Moreover, there are a variety of different criteria defining a metabolically healthy status (*Alberti, Zimmet & Shaw, 2006*; *Alberti et al., 2009*; *Expert Panel on Detection E & Treatment of High Blood Cholesterol in A, 2001*). We adopted a recently published, stricter definition of MHO (*Smith, Mittendorfer & Klein, 2019*) to explore the relationship between different obesity phenotypes and Afib.

## MATERIALS & METHODS

### Study participants

The participants were recruited from the Taiwan's Hypertensive, Hyperglycemia, Hyperlipidemia Survey (TwSHHH) conducted in 2002. The sample was based on the Taiwan National Health Interview Survey (NHIS) conducted in 2001. A multi-stage, stratified systematic sampling method was adopted, and the population was a representative community-based cohort. The detailed information of the study cohorts of TwSHHH and NHIS was described in previous publications (*Hwang, Bai & Chen, 2006*).
Well-trained nurses interviewed the participants during face-to-face home visits using a uniform questionnaire to obtain sociodemographic information. The original cohort was composed of 10,292 people. The blood pressure and anthropometric measurements were performed by trained nurses. Blood pressure was measured in each participant's right arm at least twice, and the average was calculated and adopted in this study. Waist circumference (WC) was measured at the narrowest point of the waist or the middle point of the last rib margin and iliac crest. Blood was collected to examine fasting serum glucose and lipid levels. After excluding patients with incomplete questionnaires, incomplete biochemistry or anthropometric measurements, or a lack of informed consent, 6,706 people attended this program. We further excluded participants younger than 20 years old, those with missing or incomplete body mass index (BMI) data, those who had been pregnant in the last year, or those with a previously established diagnosis of Afib or atrial flutter.

The study was deemed to be exempt from IRB approval by the Institutional Review Board of National Taiwan University Hospital (201901103W).

## Definitions of different obesity phenotypes

We defined obesity using BMI, and the cutoff was adopted according to the Ministry of Health and Welfare of Taiwan (Hwang, Bai & Chen, 2006). We categorized the participants into the underweight (BMI < 18.5 kg/m2), normal weight (BMI = 18.5–23.9 kg/m2), and overweight/obesity groups (BMI ≥ 24 kg/m2). Overweight and obese subjects were combined because of their small numbers.

Metabolically healthy was defined with some modifications according to a study published in 2019 (Smith, Mittendorfer & Klein, 2019). As the metabolically healthy group, the participants had not been diagnosed with hypertension, type two diabetes mellitus, or hyperlipidemia. Hypertension was defined as systolic blood pressure ≥ 140 mmHg and diastolic blood pressure ≥ 90 mmHg or the use of more than 28 tablets of antihypertensive therapy in the past year. Diabetes mellitus was defined as fasting serum glucose ≥ 126 mg/dL and hemoglobin A1C ≥ 6.5% or the use of more than 28 tablets of antidiabetic therapy in the past year. Hyperlipidemia was defined as low-density lipoprotein ≥ 160 mg/dL or the use of more than 28 tablets of lipid-lowering therapy in the past year. In addition to the exclusion of cardiometabolic diseases, the participants met the following criteria: (1) systolic blood pressure < 130 mmHg and diastolic blood pressure < 85 mmHg, (2) fasting serum glucose < 100 mg/dL, (3) triglyceride < 150 mg/dL, and (4) high-density lipoprotein ≥ 40 mg/dL (men) or ≥ 50 mg/dL (women). Otherwise, participants were considered metabolically unhealthy.

Therefore, we had six groups in our study: metabolically healthy underweight (MHUW), metabolically healthy normal weight (MHNW), metabolically healthy overweight/obesity (MHOW/MHO), metabolically unhealthy underweight (MUUW), metabolically unhealthy normal weight (MUNW), and metabolically unhealthy overweight/obesity (MUOW/MUO) groups.

## Outcome measurements

*Via* linkage to The National Health Insurance Research Database (NHIRD), which covered 99% of Taiwan's health insurance record (*Hwang, Bai & Chen, 2006*), the participants' outpatient clinic records, inpatient data, catastrophic illness records, prescription records, and death records were retrieved.

The endpoints of this study were new-onset Afib or atrial flutter and their related mortality events. The diagnosis of Afib and atrial flutter was based on the International Classification of Diseases, Ninth Revision-Clinical Modification (ICD-9-CM) codes obtained from the NHIRD from 2001 to 2015, and the death records from Taiwan's National Death Registry were traced and confirmed. The detailed information of ICD codes is presented in Table S1.

## Statistical analysis

Statistical analysis was performed using SAS software (version 9.4; SAS Institute, Cary, NC, USA) and Stata version 15 (Stata Corporation, College Station, TX, USA), and the two-tailed test significance level was $p < 0.05$. We analyzed basic demographic data using analysis of covariance to check continuous variables, which were presented as the mean ± SD, and the chi-squared test was used for categorical variables, which were presented as frequencies and percentages.

Person-years were calculated since the beginning of the study, and the endpoint was set as the outcome occurred. If the event did not occur, data were censored on December 31, 2015, which was the end of the study. The incidence of Afib was calculated by dividing the number of cases by the sum of 1,000 person-years of follow-up.

The time-dependent covariate regression method was used to assess the proportional hazards assumption, and the test result was insignificant. The log-rank test ($p < 0.001$) was used to assess differences in survival rates among the six groups. Cox regression models were constructed to calculate the multivariable adjusted hazard ratios (HRs) and 95% confidence intervals (CIs) of the incidence of Afib for each metabolic obesity phenotype compared to the MHNW group. Possible confounding factors according to general principles and reviews of previous studies (*Feng et al., 2019*; *Huang et al., 2020*; *Kwon et al., 2019*; *Lee et al., 2017*; *Nystrom et al., 2015*; *Vyssoulis et al., 2013*) were selected. Model 1 was adjusted for sex and age; model 2 was adjusted for all model 1 variables plus current smoking status, alcohol use, and regular exercise; model 3 was adjusted for all model 2 variables plus marital status, education level, and average monthly income.

## Sensitivity analysis

(1) Abdominal obesity: we adopted WC to redefine obesity instead of BMI. According to the 2000 WHO Asia Pacific Guidelines, central obesity is defined as WC ≥ 90 cm in men and WC ≥ 80 cm in women (*World Health Organization. Regional Office for the Western, 2000*).

(2) Exclusion of first-year events: We excluded Afib and atrial flutter events in the first year of the study to avoid an inverse causal relationship during the latent period.

(3) Redefinition of metabolically healthy: To exclude diabetes, hypertension, and hyperlipidemia, we adopted the metabolic profiles measured in TwSHHH or those recorded in the pharmacy record. However, some patients may have had a diagnosis of diabetes, hypertension, or hyperlipidemia *via* linkage to the NHIRD without receiving treatment. Thus, we had to examine the definitions of diabetes, hypertension, and hyperlipidemia. We redefined metabolically unhealthy as the presence of abnormal metabolic profiles or at least two ICD-9 codes of diabetes (ICD-9 code 250), hypertension (ICD-9 codes 401–405, 437.2), or hyperlipidemia (ICD-9 code 272) in their outpatient clinic or inpatient admission records.

## RESULTS

Ultimately, our study included 5,742 participants (Fig. S1). In this study, the follow-up rate was 99.4% of that in the National Health Interview Survey (NHIS). The mean participant age was 44.1 ± 15.5 years old. Women accounted for 49.7% of the participants. The MHOW/MHO group comprised 8.6% of the study cohort (493 participants), whereas the MUNW and MUOW/MUO groups comprised 29.2% and 30.1% of the subjects, respectively.

Table 1 presents the baseline characteristics of the participants in the six groups. The mean BMI was 23.3 ± 3.6 kg/m2. There was a generally increasing trend of elevation of fasting glucose, hemoglobin A1C, systolic and diastolic blood pressure, total cholesterol, triglyceride, and low-density lipoprotein in the order of MHUW > MHNW > MHOW/MHO > MUUW > MUNW > MUOW/MUO, whereas high-density lipoprotein followed the opposite trend (Table 1).

Table 2 presents the risk of Afib among the six groups. During a median follow-up of 13.7 years, 148 patients developed Afib. Over 6,570 person-years of follow-up, nine events occurred in the MHOW/MHO group, and the incidence rate was 1.37 per 1,000 person-years. The study result indicated a numerically higher but insignificant risk (adjusted HR = 1.97, 95% CI [0.80–4.86]) of Afib in the MHOW/MHO group than in the reference group.

Concerning the metabolically unhealthy groups, after adjustment for model 3 variables, both the MUNW and MUOW/MUO phenotypes were associated with significantly higher risks of Afib, and the adjusted HRs were 2.64 (95% CI [1.34–5.17]) and 2.20 (95% CI [1.12–4.33]), respectively. However, MUUW did not significantly increase the risk of Afib (Table 2).

The results of sensitivity analysis are presented in Table 3. After redefining obesity based on WC, the metabolically healthy with abdominal obesity (HR = 1.79, 95% CI [0.66–4.86]) displayed trends toward higher risks of Afib. Metabolically unhealthy with optimal WC (HR = 2.66, 95% CI [1.38–5.15]), and metabolically unhealthy with abdominal obesity groups (HR = 2.19, 95% CI [1.14–4.22]) were associated with significantly higher risks of Afib. After excluding first-year events or redefining metabolically healthy to exclude participants with ICD codes, the MHOW/MHO, phenotype was linked to numerically higher but insignificant risks of Afib.

## DISCUSSION

To our knowledge, our study is the first representative community-based cohort study exploring the relationship between different metabolic obesity phenotypes and Afib risk in
**Table 1 Baseline characteristics of participants according to the metabolic status and body mass index categories.**

| Characteristics | Overall | | Metabolically healthy (n = 2,181) | | | | | | Metabolically unhealthy (n = 3,561) | | | | | | p value |
| --- | --- | --- | --- | --- | --- | --- | --- | --- | --- | --- | --- | --- | --- | --- | --- |
| | | | UW | | NW | | OW/OB | | UW | | NW | | OW/OB | | |
| | n | % | n | % | n | % | n | % | n | % | n | % | n | % | |
| Patients | 5,742 | 100.0 | 205 | 3.6 | 1,483 | 25.8 | 493 | 8.6 | 156 | 2.7 | 1,678 | 29.2 | 1,727 | 30.1 | |
| 20–39 (years old) | 2,441 | 42.5 | 160 | 78.1 | 862 | 58.1 | 197 | 40.0 | 91 | 58.3 | 638 | 38.0 | 493 | 28.6 | <0.0001 |
| 40–64 (years old) | 2,602 | 45.3 | 40 | 19.5 | 559 | 37.7 | 265 | 53.8 | 37 | 23.7 | 754 | 44.9 | 947 | 54.8 | <0.0001 |
| ≥65 (years old) | 699 | 12.2 | 5 | 2.4 | 62 | 4.2 | 31 | 6.3 | 28 | 18.0 | 286 | 17.0 | 287 | 16.6 | <0.0001 |
| Women | 2,851 | 49.7 | 163 | 79.5 | 853 | 57.5 | 231 | 46.9 | 101 | 64.7 | 822 | 49.0 | 681 | 39.4 | <0.0001 |
| Current smokers | 1,378 | 24.0 | 28 | 13.7 | 297 | 20.0 | 94 | 19.1 | 33 | 21.2 | 438 | 26.1 | 488 | 28.3 | <0.0001 |
| Alcohol used | 1,604 | 27.9 | 32 | 15.6 | 358 | 24.1 | 145 | 29.4 | 27 | 17.3 | 474 | 28.3 | 568 | 32.9 | <0.0001 |
| Regular exercise habit | 1,357 | 23.6 | 34 | 16.6 | 343 | 23.1 | 131 | 26.6 | 27 | 17.3 | 397 | 23.7 | 425 | 24.6 | 0.0276 |
| Living with spouse | 3,721 | 64.8 | 84 | 41.0 | 883 | 59.5 | 356 | 72.2 | 57 | 36.5 | 1,060 | 63.2 | 1,281 | 74.2 | <0.0001 |
| Educational level | 3,238 | 56.4 | 165 | 80.5 | 1,019 | 68.7 | 278 | 56.4 | 99 | 63.9 | 911 | 54.3 | 766 | 44.4 | <0.0001 |
| Income (NTD) | 1,218 | 21.2 | 29 | 14.2 | 307 | 20.7 | 138 | 28.0 | 15 | 9.6 | 343 | 20.4 | 386 | 22.4 | <0.0001 |
| | Mean | SD | Mean | SD | Mean | SD | Mean | SD | Mean | SD | Mean | SD | Mean | SD | |
| Age (years old) | 44.1 | 15.5 | 32.7 | 11.8 | 38.1 | 12.9 | 43.4 | 13.1 | 40.7 | 19.8 | 46.3 | 16.7 | 48.9 | 14.6 | <0.0001 |
| BMI (kg/m$^2$) | 23.3 | 3.6 | 17.6 | 0.8 | 21.2 | 1.5 | 26.4 | 2.9 | 17.7 | 0.7 | 21.7 | 1.5 | 27.0 | 2.7 | <0.0001 |
| WC (cm) | 80.5 | 10.9 | 64.5 | 4.9 | 73.4 | 6.7 | 86.0 | 8.2 | 69.3 | 7.4 | 78.2 | 7.7 | 90.1 | 8.7 | <0.0001 |
| SBP (mmHg) | 116.0 | 17.8 | 101.6 | 10.0 | 105.6 | 10.0 | 110.3 | 9.6 | 110.2 | 19.1 | 120.5 | 19.4 | 125.7 | 17.7 | <0.0001 |
| DBP (mmHg) | 75.5 | 11.4 | 66.5 | 7.9 | 69.2 | 7.6 | 72.9 | 6.8 | 71.2 | 11.9 | 77.3 | 11.5 | 81.9 | 11.4 | <0.0001 |
| FPG (mg/dL) | 94.5 | 28.7 | 83.6 | 7.0 | 85.0 | 6.7 | 87.6 | 6.5 | 95.4 | 31.4 | 98.6 | 34.6 | 104.2 | 37.4 | <0.0001 |
| HbA1c (%) | 5.37 | 1.05 | 4.98 | 0.55 | 5.03 | 0.47 | 5.17 | 0.51 | 5.21 | 0.96 | 5.48 | 1.30 | 5.74 | 1.28 | <0.0001 |
| TCHO (mg/dL) | 185.9 | 37.6 | 171.0 | 25.9 | 176.2 | 28.1 | 183.0 | 27.0 | 168.2 | 38.9 | 187.7 | 41.8 | 198.1 | 42.1 | <0.0001 |
| Triglycerides (mg/dL) | 129.2 | 85.4 | 74.1 | 24.1 | 83.5 | 26.8 | 95.5 | 27.8 | 107.5 | 65.4 | 142.5 | 84.3 | 184.6 | 106.1 | <0.0001 |
| HDL (mg/dL) | 55.6 | 15.2 | 63.5 | 11.4 | 61.5 | 11.6 | 60.8 | 11.6 | 53.7 | 13.4 | 52.6 | 16.3 | 49.5 | 15.8 | <0.0001 |
| LDL (mg/dL) | 116.8 | 27.0 | 101.3 | 18.4 | 107.3 | 20.6 | 113.7 | 20.2 | 105.0 | 27.5 | 120.3 | 29.3 | 127.3 | 28.7 | <0.0001 |

Notes:

Current smokers: smoking in the recent 1 month; Alcohol use: The answer to 'Do you drink alcohol?' is 'yes'; Regular exercise habit: at least three times a week for at least 30 min for a period of more than 3 months; Living with spouse: Married, and living with a spouse, cohabitation; Educational level: ≥9 years of schooling; Income: Average month income ≥ 40,000 (New Taiwanese Dollar).

Body mass index categories: Underweight, <18.5 kg/m$^2$; Normal weight, 18.5 to 23.9 kg/m$^2$; Overweight, 24.0 to 26.9 kg/m$^2$; obese, ≥27.0 kg/m$^2$.

Abbreviations: UW, underweight; NW, normal weight; OW, overweight; OB, obesity; NTD, New Taiwanese Dollars; BMI, body mass index; WC, Waist circumference; SBP, Systolic blood pressure; DBP, Diastolic blood pressure; FPG, Fasting plasma glucose; TCHO, Total cholesterol; HbA1c, Hemoglobin A1C; HDL, High-density lipoprotein cholesterol; LDL, Low-density lipoprotein cholesterol.

Taiwan. Our study revealed that the MHOW/MHO, MUNW, and MUOW/MUO phenotypes may carry higher risks of future Afib than the MHNW phenotype. Meanwhile, in all sensitivity tests, the trends of increased Afib risks among the MHOW/MHO, MUNW, and MUOW/MUO phenotypes remained unchanged.

## Prevalence of metabolic and obesity phenotypes

Previous studies reported the global overall prevalence of MHO as 7.27–35% (*Wang et al., 2015*) with significant regional differences. A study involving Asian patients (*Lee, 2009*) reported a prevalence of MHO of 15.2% using a cutoff of BMI ≥ 25 kg/m$^2$. However, the prevalence of MHOW/MHO was only 8.6% in our study.

**Table 2 The risk of atrial fibrillation incidence according to the metabolic status and body mass index categories.**

| Variables | Metabolically healthy | | | Metabolically unhealthy | |
| --- | --- | --- | --- | --- | --- |
| | Underweight | Normal weight | Overweight/obese | Underweight | Normal weight |
| Participants | 205 | 1483 | 493 | 156 | 1,678 |
| Person-years | 2734.23 | 19870.66 | 6570.05 | 1,932.74 | 2,1281.12 |
| Events | 1 | 10 | 9 | 1 | 65 |
| Incidence rate | 0.37 | 0.50 | 1.37 | 0.52 | 3.05 |
| **Hazard ratio (95% CI)** | | | | | |
| Model 1[a] | 1.34 [0.17–10.46] | 1 (reference) | 2.04 [0.83–5.02] | 0.61 [0.08–4.81] | 2.70 [1.38, 5.29]* |
| Model 2[b] | 1.33 [0.17–10.45] | 1 (reference) | 2.00 [0.81–4.93] | 0.60 [0.08–4.74] | 2.70 [1.38, 5.29]* |
| Model 3[c] | 1.30 [0.17–10.20] | 1 (reference) | 1.97 [0.80–4.86] | 0.60 [0.08–4.68] | 2.64 [1.34, 5.17]* |

Notes:
Incidence rates are presented per 1,000 person-years; *Data are statistically significant ($p < 0.05$).
Body mass index categories: Underweight, <18.5 kg/m$^2$; Normal weight, 18.5 to 23.9 kg/m$^2$; Overweight, 24.0 to 26.9 kg/m$^2$; obese, ≥27.0 kg/m$^2$.
Abbreviations: CI, confidence interval.
a, adjusted for sex, age; b, adjusted for sex, age, smoker, alcohol, regular exercise habit; c, adjusted for sex, age, smoker, alcohol, regular exercise habit, marital status, education, income.
Covariates: Current smokers: smoking in the recent 1 month; Alcohol use: The answer to 'Do you drink alcohol?' is 'yes'; Regular exercise habit: at least three times a week for at least 30 min for a period of more than 3 months; Living with spouse: Married, and living with a spouse, cohabitation; Educational level: ≥9 years of schooling; Income: Average month income ≥40,000 (New Taiwanese Dollar).

**Table 3 Sensitivity analyses of the risk of atrial fibrillation incidence according to the metabolic status and body mass index categories.**

| | Metabolically healthy | | | Metabolically unhealthy | | |
| --- | --- | --- | --- | --- | --- | --- |
| | | Optimal | Abdominal obesity | | Optimal | Abdominal obesity |
| Obesity defined by WC | | 1 | 1.79 (0.66, 4.86) | | 2.66 (1.38, 5.15)* | 2.19 (1.14, 4.22)* |
| | Metabolically healthy | | | Metabolically unhealthy | | |
| | UW | NW | OW/OB | UW | NW | OW/OB |
| Excluding events in the first year | 1.43 (0.18, 11.36) | 1 | 1.67 (0.62, 4.50) | 0.71 (0.09, 5.64) | 2.76 (1.35, 5.61)* | 2.33 (1.14, 4.74)* |
| Re-definition of metabolically healthy | 1.33 (0.17, 10.46) | 1 | 1.64 (0.62, 4.33) | 0.47 (0.06, 3.70) | 2.12 (1.07, 4.18)* | 1.88 (0.95, 3.70) |

Notes:
Adjusted by model three, presented with hazard ratio with 95% confidence interval. *Data are statistically significant ($p < 0.05$).
Model 3: adjusted for gender, age, smoker, alcohol, marital status, education, income.
Optimal waist circumference: <80 cm in women and <90 cm in men; abdominal obesity, ≥80 cm in women and ≥90 cm in men; Underweight, <18.5 kg/m$^2$; Normal weight, 18.5 to 23.9 kg/m$^2$; Overweight, 24.0 to 26.9 kg/m$^2$; obese, ≥27.0 kg/m$^2$.
Re-definition of metabolically healthy: Exclusion with abnormal metabolic profiles in the TwSHHH survey or with at least twice the diagnosis of International Classification of Diseases-9th revision codes of diabetes, hypertension, and hyperlipidemia in the outpatient clinic or the inpatient admission record.
Abbreviations: UW, underweight; NW, normal weight; OW, overweight; OB, obesity; WC, Waist circumference. TwSHHH, Taiwan's Hypertensive, Hyperglycemia, Hyperlipidemia Survey.

In our study, the prevalence of MUNW was 29.2%, and the MUOW/MUO group even accounted for 30.1% participants. A previous study reported rates of MUNW and MUO were 19.98 and 17.91%, respectively (*Wang et al., 2015*).

The different rates of metabolic and obesity phenotypes may lie in inconsistent criteria defining a metabolically healthy status in each study. Our study adopted a stricter criterion to define metabolically healthy as rigorously as possible. Hence, the prevalence of a metabolically healthy status was lower and that of a metabolically unhealthy status was higher than those previously reported.

## Metabolically unhealthy and Afib

In our study, both the MUNW and MUOW/MUO phenotypes were linked to significantly higher risks of future Afib after adjustment for model 3 variables. Thus,

our study suggests a higher risk of Afib for both the MUNW and MUOW/MUO phenotypes.

Our study revealed a significantly increased risk of Afib for the MUNW group compared to the MHNW group. The results of a large Korean cohort study by *Lee et al. (2017)* and a study by *Feng et al. (2019)* supported our findings. However, a Swedish study by *Nystrom et al. (2015)* reported an insignificantly higher Afib risk in the MUNW group. First, there were several differences between our study groups. We included different races, which may have contributed to the different group characteristics and subsequent study result. Second, the Afib records of the study by Nystrom and co-workers were obtained from the hospital discharge registry, which only included inpatient data, raising concerns that Afib events were underestimated. Finally, the prior study had a relatively small MUNW group of only 92 people, whereas our study included 1,678 people. The insignificant result might be explained by the limited sample size because a small study group may lead to insufficient statistical power.

Our study demonstrated a significantly higher risk of Afib in the MUOW/MUO group. The study result is also supported by the other three cohort studies that reached an identical conclusion.

Although the pathogenesis of metabolic unhealthy-associated Afib is not well understood, it is believed to be a complex interaction among metabolic, mechanical, and environmental factors (*Watanabe et al., 2008*). Low high-density lipoprotein levels related to the pro-inflammatory milieu have been suggested to increase Afib susceptibility (*Ahn et al., 2021*). Thus, a mechanism related to inflammation and oxidative stress has been proposed to have a common etiology and has been implicated in the pathogenesis (*Watanabe et al., 2008*). Another possible mechanism is that mechanical stress-related atrial enlargement may lead to electrophysiologic changes and result in Afib (*Healey & Connolly, 2003*). Among the metabolic components, the relationship between hypertension and left atrial enlargement is well established (*Healey & Connolly, 2003*). Meanwhile, one study reported that MetS was associated with the atrial enlargement observed in patients with non-valvular Afib (*Nicolaou et al., 2007*). More studies are expected because of the lack of evidence regarding the pathogenesis.

## MHO and Afib

A previous cohort study demonstrated an increased risk of cardiovascular disease in the MHO group, but no Afib data were presented (*Yeh et al., 2021*). Our study reported a 1.97-fold higher risk in the MHOW/MHO group, although this result was not statistically significant. *Lee et al. (2017)*, *Feng et al. (2019)* and *Nystrom et al. (2015)* identified a significantly increased risk. However, disparities among our study groups could explain our insignificant result. First, our study group had a slightly lower BMI than those of *Feng et al. (2019)* and *Lee et al. (2017)*. Second, because of the different criteria of a metabolically healthy status, the MHO groups in the studies by *Lee et al. (2017)*, *Feng et al. (2019)* and *Nystrom et al. (2015)* might have included people with obesity and fewer metabolic components. These differences might explain the significantly increased risk of Afib in their studies. However, the trend of higher Afib risk in the MHO group even under stricter

criteria in our study was non-negligible. Meanwhile, *Feng et al. (2019)* demonstrated an insignificant but numerically higher risk of Afib for the MHOW and MHO groups with more rigorous criteria in a sensitivity test, supporting our findings.

The possible mechanisms proposed by previous researchers to describe the increased risk of Afib associated with MHO were associated with obesity-related progressive atrial structural and electrical remodeling (*Abed et al., 2013*), the accumulation of pericardial fat (*Nalliah et al., 2016*), and a systemic pro-inflammatory condition generated by cytokines released by adipose tissue (*Scridon et al., 2015*).

However, these mechanisms remained uncertain, and more studies are needed to compare the risk of Afib between among metabolically obesity phenotypes.

## Abdominal obesity and Afib

Abdominal obesity has been well recognized as an important component of a metabolically unhealthy status. In our sensitivity test, we redefined obesity using WC instead of BMI, and the result demonstrated a higher Afib risk, albeit with a wide CI, in the metabolically healthy with abdominal obesity group. *Nystrom et al. (2015)* recorded similar trends as our study. Meanwhile, *Feng et al. (2019)* revealed a significantly higher risk among patients with a metabolically healthy with abdominal obesity phenotype, and the disparities between our studies may lie in the cutoff of WC.

According to our study results, metabolic phenotypes determined using either BMI or WC revealed identical trends of numerically higher risks among the MHO, MUNW, and MUO phenotypes. Thus, both BMI and WC could be evaluated in the context of metabolic obesity and Afib.

## Study strengths and limitations

Our study applied a more precise definition, and we attempted to identify the target group of truly metabolically healthy people and avoid an underestimation of the metabolically unhealthy population. With its representative, community-based structure, this study demonstrated the relationships of Afib with metabolic obesity phenotypes. However, our study had several limitations. This was a prospective cohort study without randomization. Under the observational structure, we could not establish clear causal relationships in our study. In addition, silent Afib was not traced because remote heart monitoring was not employed. Thus, asymptomatic Afib incidents may have been missed in the absence of events necessitating an insurance claim, which could have led to underestimation of the Afib incidence in our study. Additionally, although we have adjusted for several confounding factors, other residual covariates such as medication history may also need to be considered. Another limitation is that in our study, there were still not enough information about the precise physical activity content because there were a variety of types, frequency, and strengths of physical activity. Besides, fitness may have an impact on MHO. It is a potential factor worth analyzing. We all agree with the value of a more detailed analysis on physical activity and fitness in the future. Finally, metabolic obesity phenotypes may change over time, and we did not analyze the effect of phenotype

transition. Future studies should consider metabolic obesity phenotype transition and the resulting effect on Afib risk.

## CONCLUSIONS

This was the first representative community-based cohort study exploring the relationships of different metabolic and obesity phenotypes with Afib risk in Taiwan. Our results suggest that the risk of Afib was elevated in the MUNW, MUOW/MUO, and MHO groups. The study result indicated that the Afib risk among metabolic and obesity phenotypes is associated with a metabolically unhealthy status. Obesity itself even without other metabolic components might also be associated with the Afib risk. It will be important to manage metabolic components and obesity to alleviate the risk of Afib, and the preventive strategy for Afib should include lifestyle modification. However, the result was not robust and it still suggested further study.

## ACKNOWLEDGEMENTS

We would like to thank to Enago academy for language editing.

### Funding

The authors received no funding for this work.

### Competing Interests

The authors declare that they have no competing interests.

### Author Contributions

- Szu-Ying Tsai performed the experiments, prepared figures and/or tables, authored or reviewed drafts of the paper, and approved the final draft.
- Hsin-Hao Chen performed the experiments, prepared figures and/or tables, and approved the final draft.
- Hsin-Yin Hsu analyzed the data, authored or reviewed drafts of the paper, and approved the final draft.
- Ming-Chieh Tsai analyzed the data, authored or reviewed drafts of the paper, and approved the final draft.
- Le-Yin Hsu analyzed the data, authored or reviewed drafts of the paper, and approved the final draft.
- Lee-Ching Hwang performed the experiments, authored or reviewed drafts of the paper, and approved the final draft.
- Kuo-Liong Chien performed the experiments, authored or reviewed drafts of the paper, and approved the final draft.
- Chien-Ju Lin performed the experiments, prepared figures and/or tables, authored or reviewed drafts of the paper, and approved the final draft.
- Tzu-Lin Yeh conceived and designed the experiments, prepared figures and/or tables, authored or reviewed drafts of the paper, and approved the final draft.

## Human Ethics

The following information was supplied relating to ethical approvals (*i.e.*, approving body and any reference numbers):

The ethics review was approved by the Institutional Review Board of National Taiwan University Hospital (201901103W).

## Data Availability

The datasets generated and/or analyzed during the current study are not publicly available due to the terms of consent to which the participants agreed. The data are available from the authors upon reasonable request and with permission of the Health Promotion Administration at the Ministry of Health and Welfare in Taiwan: https://dep. mohw.gov.tw/DOS/cp-2516-3591-113.html. The approved number was H108044: https://dep.mohw.gov.tw/DOS/cp-2499-45896-113.html. We provide an institutional, non-author point of contact, ntuepm@ntu.edu.tw, where all interested researchers can direct data inquiries. Data are available from the National Taiwan University Hospital Institutional Data Access/Ethics Committee (contact *via* vdgntuepm@ntu.edu.tw) for researchers who meet the criteria for access to confidential data.

## Supplemental Information

Supplemental information for this article can be found online at http://dx.doi.org/10.7717/ peerj.12342#supplemental-information.

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
