# Peer review of "Obesity phenotypes and their relationships with atrial fibrillation"

_PeerJ, doi:10.7717/peerj.12342_

## Round 0.1 · original submission · Major Revisions

The reviewers have found scientific merit in your work. However, they have indicated some changes which you should address in a revised version of the text.

Reviewer 1 has suggested that you cite specific references. You are welcome to add it/them if you believe they are relevant. However, you are not required to include these citations, and if you do not include them, this will not influence my decision.

Reviewer 1 ·

Basic reporting

.

Experimental design

.

Validity of the findings

.

Additional comments

The paper is solid on an important topic , so this should be publishable in a solid Journal. A limitation that should be mentioned is only minimal data on physical activity (PA) and none on Fitness , and Fitness may be more important than weight ( Barry VW et al. Prog Cardiovasc Dis 2018; 61: 136-141) and even PA/Fitness even may markedly impact the MHO ( Ortega FB et al. PCVD 2018; 61: 190-205.) The authors adjusted for WC and showed their data by WC-They could combine both high WC and high TGs , as this is a group who has more marked visceral adiposity. Also, they could consider including 2 highly cited major JACC State of the Arts , one on Obesity and AF ( Lavie CJ et al. JACC 2017; 70: 2022-2035) and one on the whole idea of Healthy Weight including MHO( Lavie CJ et al. JACC 2018; 72: 1506-1531.)

Reviewer 2 ·

Basic reporting

Tsai et al. reported a normative study to explore the relationships between obesity phenotypes and atrial fibrillation. The background was well organized, and the prior literatures were approprately referenced. The results found that the risk of Afib was elevated in the MUNW and MUOW/MUO groups (without statistical significance), but these findings lack of innovation. The major hypothese about MHO and atrial fibrillation was Still unproven. The authors should derectly point that whether people with MHO would have higher risk of Afib is unknown, and need further studies.

Experimental design

This cohort was well designed, and the outcome assessment was credible. But there are some deficiencies.
1. According to the Inclusion criteria, there were 4000+ people excluded because of various reasons, please discuss possible selection bias from this progress.
2. Line 95, why you further excluded participants younger than 20 years old?
3. Suggest to add a flow chart figure to show the participants selection, and to report the rate of lost to follow-up, these is important for cohort study.
4. Line 142: the author used "The time-dependent covariate regression method was used to assess the proportional hazards assumption", what is the results? Whether all the Covariates meet the "proportional hazards assumption"? If not, how do you solve this problem?
5. The event count is so small in some groups (From Table 2), I doubt that whether the Cox regression model is suitable for HR estimations. Please demonstrate its applicability. Whether possion regreesion is more suitable?

Validity of the findings

1. From the Abstract, all the results you stated were Not statistically significant, but the conclusion is "manage metabolic components and obesity to mitigate the risk
of Afib". It is inappropriate. The results were not robust, it suggested futher studies.
2. The authors should think more carefully about Model C and Model D. The valvular heart disease, myocardial infarction, and heart failure are all associated with atrial fibrillation? All of these covarites were added to Cox model, and the HRs were weakened (insignificant). How to explain these results? you should consider clinical and statistical correctness.
3. The authors selectively reported the results about MUNW, MUOW/MUO, and MHO groups, even though the HRs were without statistical significance. Why not explain the results from "Metabolically unhealthy Underweight" group? It seems intersting

Additional comments

1. Some typo, eg Line 254, "he"?

---

## Round 0.2 · accepted · Accept

All the reviewers' concerns have been correctly addressed.

Reviewer 2 ·

Basic reporting

The authors have adequately addressed my concerns.

Experimental design

None.

Validity of the findings

The authors have addressed the limitations that were pointed out and added the appropriate disclaimers to the findings.